# An insect-scale artificial visual-olfactory bionic compound eye

Jiachuang Wang[1,2,3], Shuai Wei[1,2], Nan Qin[1,2] ✉ & Tiger H. Tao [1,2,3,4,5] ✉

Compound eyes feature unique optical structures and high-efficiency image processing. The opto-olfactory nervous system of *Drosophila* has the characteristics of lightweight and low power consumption. Significant efforts have been dedicated to the design and manufacturing of artificial compound eye system. However, it is still challenging to construct a bionic visual-olfactory compound eye microsystem with sensitive photoelectric response and accurate olfactory perception in insect-scale, mimicking the biological multimodal fusion decision-making mechanism. Here, we report a miniature apposition compound eye that integrates 1027 ommatidia on 1.5×1.5 mm² by manufacturing a bionic micro-lens array onto flexible photodetectors via femtosecond laser two-photon polymerization, further construct the colorimetric olfactory sensor array through inkjet printing to achieve integrated perception of vision and smell. The bionic compound eye (bio-CE) enables wide field-of-view imaging (azimuth angle 180°), natural interocular isolation, a 1 kHz flicker fusion frequency and color response to various hazardous chemicals, resulting in high sensitivity to moving objects and rapid response to environmental gases. The microsystem can serve as a wide-angle close-range obstacle avoidance detector and a device for monitoring visual and olfactory information of moving targets. The insect-scale bionic apposition compound eye shows great potential applications in unmanned platform navigation and bionic robot intelligence.

The bionic compound eye is a delicate and complex visual system found in arthropods such as crustaceans and insects for dexterous hunting activities and obstacle avoidance[1,2] However, each ommatidium has a limited receptive angle which can only detect light signals in a specific direction[3–5]. Submillimeter compound eyes achieve large-scale visual imaging by integrating hundreds of micro-scale ommatidia. Such miniature photo-detecting system features wide field-of-view (FOV), distortion-free imaging, and excellent motion sensitivity[6].

Over the past few decades, significant efforts have been dedicated to the design and manufacturing of artificial compound eye systems using two major strategies. On the one hand, with the development of device manufacturing technology, the combination of micro-lens array with a commercially available charge-coupled device (CCD) or a complementary metal oxide semiconductor transistor (CMOS) sensor has demonstrated the feasibility for obtaining high-quality images[7–12]. However, these artificial compound eyes need onerous optical design and advanced three-dimensional processing methods to correct the uniformity of image quality between ommatidia in different positions, limiting the fast and efficient extraction of target parameters[13,14] On the other hand, recent advances of flexible photodetector arrays based on

[1]State Key Laboratory of Transducer Technology, Shanghai Institute of Microsystem and Information Technology, Chinese Academy of Sciences, Shanghai, China. [2]School of Graduate Study, University of Chinese Academy of Sciences, Beijing, China. [3]2020 X-Lab, Shanghai Institute of Microsystem and Information Technology, Chinese Academy of Sciences, Shanghai, China. [4]Center of Materials Science and Optoelectronics Engineering, University of Chinese Academy of Sciences, Beijing, China. [5]School of Physical Science and Technology, ShanghaiTech University, Shanghai, China. ✉e-mail: qinnan@mail.sim.ac.cn; tiger@mail.sim.ac.cn

stretchable electronics[15–23] and two-dimensional materials[24–26] has dramatically promoted the bionic compound eye research. However, limited by the fabrication process and lens-pixel assembling method, the physical size of these devices is much larger than that of arthropod compound eyes, which cannot realize a natural imaging form close to biological compound eyes[27,28] It is still a great challenge to design and fabricate an artificial compound eye with the eye scale, ommatidia density, and imaging patterns as arthropods.

Moreover, the olfactory circuit of *Drosophila* generates a specific label for each gas, which is crucial for quickly learning and responding to different odors[29,30] Research indicates that *Drosophila*-inspired locally sensitive hashing algorithm is expected to efficiently handle gas type recognition and odor concentration prediction problems[31]. Therefore, the innovation of bionic visual-olfactory integrated compound eye devices and systems could overcome the limitations of traditional imaging and processing methods, the visual-olfactory collaborative decision-making mechanism can enhance the memory effect of a single sensory signal, showing great potential in panoramic imaging, unmanned clusters, and bionic robot vision[32,33].

In this paper, we report a bionic compound eye (bio-CE) integrated on a cylindrical surface of 0.6 mm radius, which enables large-FOV imaging and moving object detection. The optoelectronic system is fabricated by directly assembling a micro-lens array on a flexible organic photodetector array via femtosecond laser two-photon polymerization (FL-TPP) and further adopt inkjet printing technology to achieve high-density array integration of color developing materials. The short focal length and small receiving angle design of the micro-lens array prevent optical interference from adjacent pixels, mimicking the natural interocular isolation in apposition compound eyes. The designed independent bionic bristle structure significantly enhances the anti-fog function and robustness of the visual imaging system in high humidity environments. Furthermore, the P3HT/PCBM/PbS-based photodetector array fulfills 0.1 ms photoelectric response and performance stability under bending conditions, the olfactory colorimetric array composed of metal complexes and pH indicators achieve a rapid gas response; the as-obtained bio-CE can be applied on unmanned platforms to demonstrate bionic object detection and obstacle avoidance, as well as environmental perception and exploration based on visual-olfactory fusion information.

## Results

### Bioinspired design of visual-olfactory bio-CE system

Among various biological vision systems[34–38], the *Drosophila* optic and olfactory nervous systems exhibit characteristics of lightweight architecture and low energy consumption. Through natural selection and evolutionary adaptation, *Drosophila* has developed a unique cross-modal integration mechanism, enabling rapid stress decision-making in complex environments. This biological paradigm offers significant potential for computational modeling and biomimetic applications. Specifically, the visual nervous system of *Drosophila* regulates flight navigation and predator avoidance, whereas the olfactory nervous system guides foraging and courtship behaviors. These visual and olfactory pathways ultimately converge in the central nervous system (CNS), where sensory inputs are processed and integrated with prior learning and cognitive experiences. This neural convergence underpins the cross-modal collaborative mechanism, allowing *Drosophila* to generate adaptive behavioral responses.

*Drosophila* possesses a compound eye visual system, characterized by highlight sensitivity and a wide field of view. Visual stimulation is achieved through beam convergence and angular constraint by the compound eye. Striated muscles function as optical waveguides, directing light to photoreceptor cells, while surrounding pigment cells provide optical isolation between individual ommatidia. In our bio-CE system, this structure is mimicked using a bionic microlens array, which converges, propagates, and optically isolates incident light beams. A photodetector array then converts the target's optical signals into electrical responses. Compared to primates, *Drosophila*'s olfactory nervous system exhibits exceptional sensitivity. Olfactory stimuli are detected by antennal olfactory receptors and processed by olfactory sensory neurons (OSNs). Similarly, in bio-CE systems, an olfactory colorimetric array enables gas recognition and concentration visualization via specific colorimetric reactions (Fig. 1A).

Inspired by the *Drosophila* compound eye (Fig. 1B), we fabricated a bioinspired imaging system (Supplementary Note 1) comprising a microlens array (MLA) (Fig. 1C) and a flexible organic photodetector (OPD) array (Fig. 1D). Each microlens was precisely aligned with an individual OPD pixel via MEMS-guided two-photon polymerization (TPP) (Supplementary Note 2). Notably, the polydimethylsiloxane (PDMS) film serves dual roles: as both an encapsulation layer for the OPD array and a transparent substrate/optical medium for the MLA. The resulting flexible bio-CE (Fig. 1E) integrates 1027 discrete ommatidia within a $1.5 \times 1.5$ mm$^2$ area, demonstrating exceptional mechanical compliance and conformal surface adaptability. This design successfully emulates the structural morphology of *Erbenochile erbeni*[39,40] while achieving a wide field-of-view imaging capability that bridges biological inspiration with advanced microfabrication technology for applications in compact vision systems and bioelectronic devices.

In contrast to superposition compound eyes[41,42] and existing bionic counterparts[13,43] our bio-CE exhibits two distinctive operational principles: (1) A unique single-pixel-per-lens architecture where targets project as intensity-modulated diffuse spots across the imaging plane. This configuration enables exceptional motion sensitivity, as positional changes induce detectable flickering at imaging zone peripheries (Supplementary Fig. 4). (2) Unlike conventional CMOS-based designs requiring full-image capture per ommatidium, our system's sparse data acquisition enables significantly higher frame rates and enhanced temporal resolution through reduced computational overhead. Furthermore, we implemented a bionic self-cleaning mechanism featuring inter-lens micro-setae arrays (Supplementary Fig. 5), structurally analogous to *Drosophila* ocular hairs. As demonstrated by Guillermo J. Amador[44,45], these densely packed microstructures effectively dampen near-surface airflow, thereby minimizing droplet deposition on optical surfaces under high-humidity conditions and maintaining consistent ommatidial performance.

The spatial resolution of our bio-CE is lower than high-density CMOS sensors. This is an inherent trade-off of the sparse sampling architecture, which prioritizes wide-field motion detection over static image detail. However, our bio-CE achieves >1000 Hz effective motion sampling. This aligns with bio-inspired vision goals where speed and energy efficiency are critical (e.g., drone navigation, robotic collision avoidance). The resolution-sensitivity trade-off is intentional for target applications. Inspired by the obstacle avoidance capabilities of *Drosophila*, we present a bio-inspired compound eye activation model (Supplementary Note 3) that achieves high-sensitivity motion detection through low-resolution image processing. This biomimetic approach demonstrates how efficient motion perception can be realized with minimal computational resources, mirroring the energy-efficient visual processing observed in insect vision systems. In summary, while the bio-CE's spatial resolution is lower than CMOS, its motion detection capabilities (speed, sensitivity, power efficiency) are superior for dynamic vision tasks.

The highly efficient neural pathways governing visual and olfactory perception in *Drosophila* have attracted considerable scientific interest. In the visual pathway, stimuli are first detected by the compound eyes and transduced by photoreceptor cells. These signals then propagate sequentially through the lamina, medulla, and lobula complex within the optic lobe, where event-level feature extraction occurs, before converging on the central complex and mushroom body in the

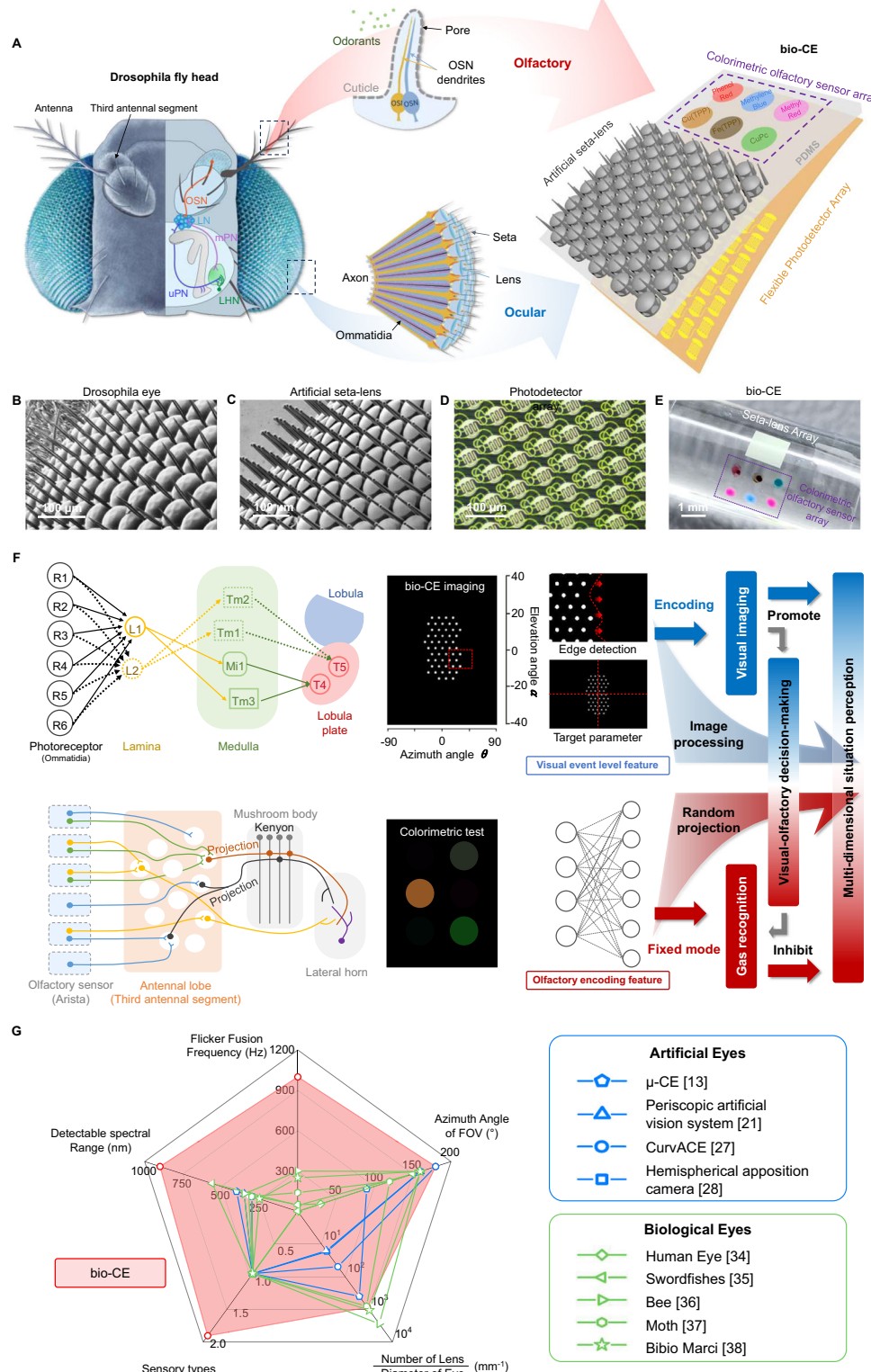

**Fig. 1 | Design of the bionic compound eye (bio-CE) system. A** The bionic correspondence between the working principle of neurons and cell levels in *Drosophila* compound eye and artificial visual-olfactory fusion compound eye system. The bio-CE system consists of artificial micro seta-lens, flexible photodetector array, and colorimetric olfactory sensor array; **B** Scanning electron microscopy (SEM) photograph of zoom in *Drosophila* eye, scale bar 100 μm; **C** SEM image of the lens-setae structure on a flexible micro-lens array, scale bar 100 μm; **D** Optical image of the organic photodetector array, scale bar 100 μm; **E** Optical image of curved bio-CE device, scale bar 1 mm; **F** The structure of visual and olfactory neural circuits in *Drosophila* and the corresponding visual-olfactory fusion algorithm is designed to achieve multi-dimensional situation perception by drawing on the cross-modal learning mechanism of *Drosophila*; **G** Comparison chart of this work (bio-CE) with other artificial eyes and biological eyes in terms of fusion frequency, azimuth angle, lens density, sensory type, and measurable spectral range.

central brain for perceptual decision-making. In the olfactory pathway, odorants are sensed by olfactory receptor neurons (ORNs) and relayed to projection neurons (PNs) in a convergent manner. Following initial integration, these signals undergo sparse random projection onto Kenyon cells (KCs) within the mushroom body, where they are modulated by inhibitory input from anterior paired lateral (APL) neurons. Ultimately, processed olfactory information is transmitted to local neurons in the lateral horn (LH) for classification and recognition. Building upon these principles, we developed an efficient visual imaging target parameter extraction method capable of deriving target position and motion state data from frame-by-frame image variations. Additionally, we designed a lightweight local sensitive hash neural network algorithm to estimate target gas types and concentrations based on stable olfactory-color responses. Furthermore, inspired by the shared neural architecture for multisensory integration in *Drosophila*, we implemented a collaborative decision-making algorithm to enhance visuo-olfactory signal fusion (Fig. 1F). Benchmarking against state-of-the-art systems, our platform demonstrates superior performance metrics across three critical parameters: detectable spectral range, flicker fusion frequency, and multimodal sensing capabilities. Notably, the system achieves biological-level performance in azimuth detection range and microlens integration density, while its front-end device architecture exhibits scale and performance characteristics comparable to natural arthropod compound eye systems (Fig. 1G).

## Design and characterization of bionic micro-lens arrays

To achieve biological compound eye-like imaging characteristics, our bio-CE system employs a bendable microlens array with three distinctive features compared to conventional planar designs: (1) a cylindrically curved configuration where each microlens maintains conformal contact with the flexible substrate; (2) precise one-to-one alignment with underlying organic photodetector pixels, exhibiting only minimal misalignment upon bending; (3) individual microlenses with narrow acceptance angles that provide effective optical isolation between adjacent ommatidia. We fabricated this bio-CE microlens array using FL-TPP technology directly on a flexible PDMS substrate (Fig. 2A), ensuring accurate alignment with the photodetector array. The microlenses are arranged in a hexagonal close-packed configuration, with their optimized geometric design enabling both superior flexibility and precise conformity to cylindrical surfaces. The acceptance angle () of the bio-CE ommatidia is determined by the focal length () and effective diameter () of the photodetector pixel according to Eq. 1.1. By precisely controlling the organic photodetector pixel area, we established a fixed acceptance angle of 16.9° to achieve imaging characteristics comparable to biological compound eyes observed in *Drosophila* and moths. In addition to analyzing individual ommatidium channels, we developed a comprehensive imaging model characterizing the complete ommatidial surface (Supplementary Note 3), which enables quantitative assessment of inter-ommatidial signal correlation, system-level analysis of compound eye imaging performance, and accurate prediction of visual field coverage and resolution. The spatial position-compound eye activation model (Fig. 2B) correlates dynamic ommatidial activation patterns with target motion parameters through quantitative analysis of activated region area, centroid coordinates, and angular distribution, allowing precise determination of target motion characteristics including relative distance, directionality, and angular position. The microlens-patterned mask imaging results (Fig. 2C) demonstrate excellent optical performance, with precise alignment between each microlens and its corresponding OPD pixel, thereby validating both the optical design specifications and the fabrication process's ability to maintain micro-scale precision.

$$\Delta\rho = 2 \arctan\left(\frac{d}{2f}\right) \qquad (1.1)$$

Notably, we engineered bionic setae structures at each triple-junction of microlenses to address humidity-related challenges. These microstructures enhance localized water vapor capture compared to planar interfaces, effectively suppressing condensation on optical surfaces under high-humidity conditions or fogging environments, thereby preserving imaging fidelity (Fig. 2D). Real-time demonstration of this anti-fogging mechanism is provided in Supplementary Movie 1.

By optimizing the surface shape and curvature of the two optical surfaces of the micro-lens, we gain effective chromatic aberration control in the visible to near-infrared spectral range (Fig. 2E) and stable convergence ability within the 0 - 10° receptive angle (Fig. 2F). Correspondingly, the design of micro-level optical structures requires higher precision processing methods. We combine the lens layer and supporting structures in an integrated model. The dome-shaped micro-lens structure can not only ensure stable support of the lens layer but also enable the developer to flow fully through the lens to obtain a high-precision optical surface. To investigate the focusing performance of the as-fabricated micro-lens, the point spread function (PSF) of a focused spot along the X-axis and Y-axis is measured under various wavelengths (Fig. 2G) and incident angle (Supplementary Fig. 6). The intensity of the focused spot shows a Gaussian distribution, indicating that the ommatidium has a high fabrication quality, and the full width at half maximum (FWHM) almost remains constant ($2.6 \pm 0.1$ μm), which suggests low aberration in the acceptive angle. Also, the height scanning results that coincide with the curvature of the optical plane can verify the accuracy and reliability of micro-lens processing (Fig. 2H).

For apposition compound eyes, optical isolation between ommatidia is achieved by the pigment cells wrapped around photoreceptor cells. In bio-CE, we conduct an ideal optical isolation effect by compressing the focal length, reducing the photosensitive surface area appropriately, and utilizing the light refraction dissipation of the lens supporting, which can avoid ghosting at large oblique incidence angles and provide the possibility to reproduce the "flicker effect" during full-frame imaging. We applied micro-lens arrays to commercial CMOS sensor (OmniVision Tech.) to test the converged energy distribution and angular response characteristics under different irradiation conditions. Figure 2I shows the normalized light intensity distribution at different incidence angles. In addition to the Gaussian spot center intensity, the convergence spots caused by the six dome supports are distributed in a hexagonal shape around the main focus, and the FWHM of the fitted angular sensitivity function (ASF) of the ommatidium is about 4.2°. Figure 2J shows the convergence of 2D spot images at different incidence angles. The yellow dotted line indicates the effective receiving area of the photodetector corresponding to the lens. With the increase of the incidence angle (0 - 20°), the convergence spot gradually deviates from the photodetector. When the incidence angle exceeds 30°, the spot energy is rapidly attenuated, indicating the optical isolation effect of the short focal length microlens array design, as detailed in Supplementary Fig. 7 and Figure 8.

## Optoelectrical performance of the organic photodetector array

The organic photodetector array was meticulously designed to complement the micro-lens array, faithfully replicating the structural and functional characteristics of columnar apposition compound eyes. As illustrated in Fig. 3A, each bio-CE ommatidium features a vertically integrated architecture comprising: a P3HT/PCBM/PbS quantum dots (QDs) photosensitive film, an SU-8 isolation layer, a gold interdigital electrode layer, a copper wiring layer, and a flexible polyimide substrate. The complete fabrication methodology and device architecture are systematically documented in Supplementary Note 1 and Figure 9, demonstrating reproducible device performance across the array. The fabricated flexible organic photodetector array demonstrates full-frame detection capability with distinct spectral responses across ultraviolet (365 nm), visible (532 nm), and near-infrared (1250 nm)

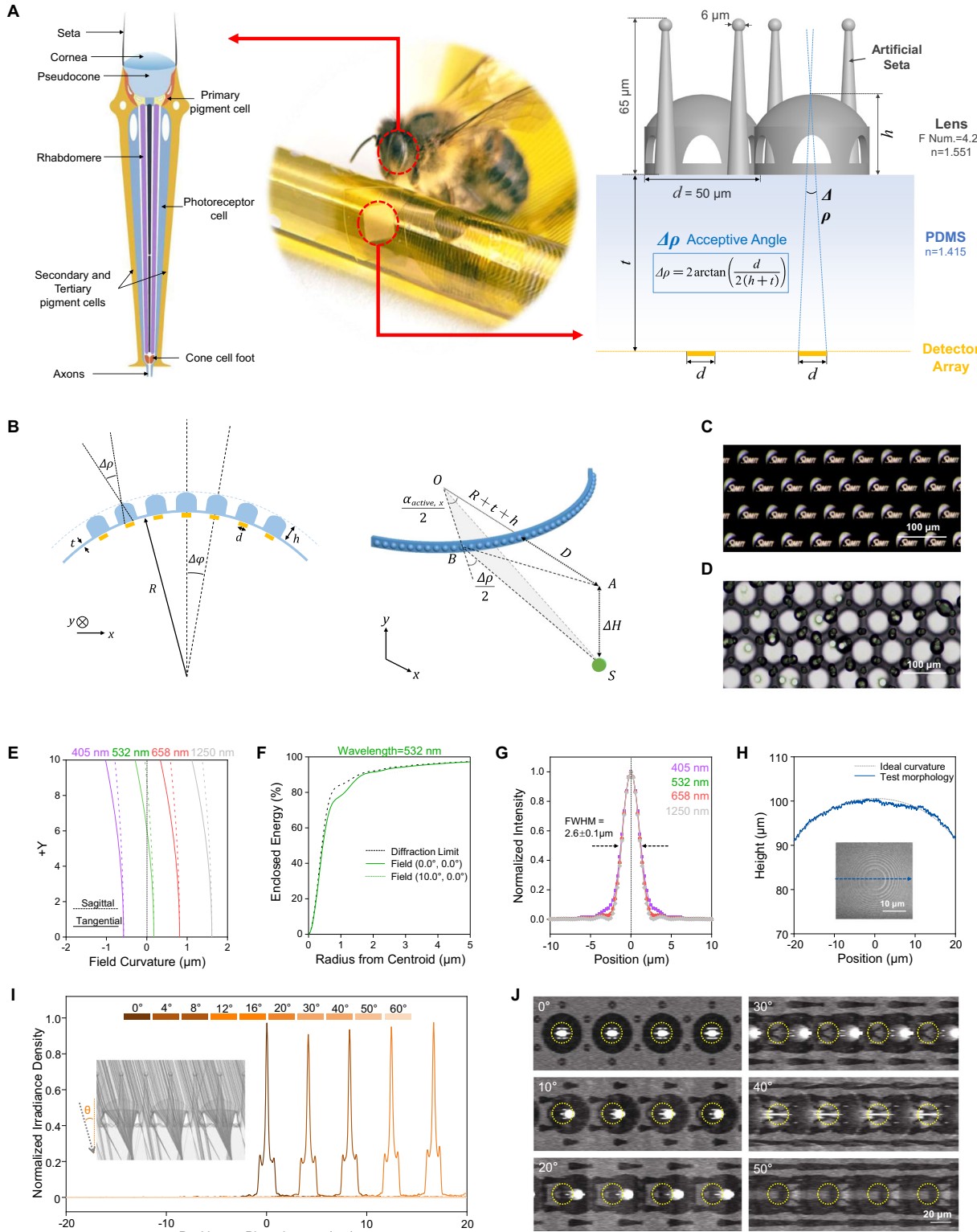

**Fig. 2 | Design, fabrication, and characterization of bionic micro-lens arrays.**
**A** The structure of *Drosophila* microphthalmia and the geometric optical parameters of bio-CE ommatidium; **B** The mathematic model of the target spatial position and the compound eye activation form; **C** Micro-lens array imaging with patterned mask, scale bar 100 μm; **D** Droplet condensation effect under high humidity condition, scale bar 100 μm; **E** Field curvature of the designed micro-lens under various wavelength illumination; **F** Encircled energy distribution curve at a different incident angle; **G** Point diffusion function of micro-lens at different wavelengths; **H** Morphology characterization of micro-lens surface, inset scale bar 10 μm; Interocular optical isolation is verified by the linear light intensity distribution of micro-lens arrays at different incidence angles (**I**) and the CMOS-based micro-lens array spot scattering varies with incidence angle (**J**).

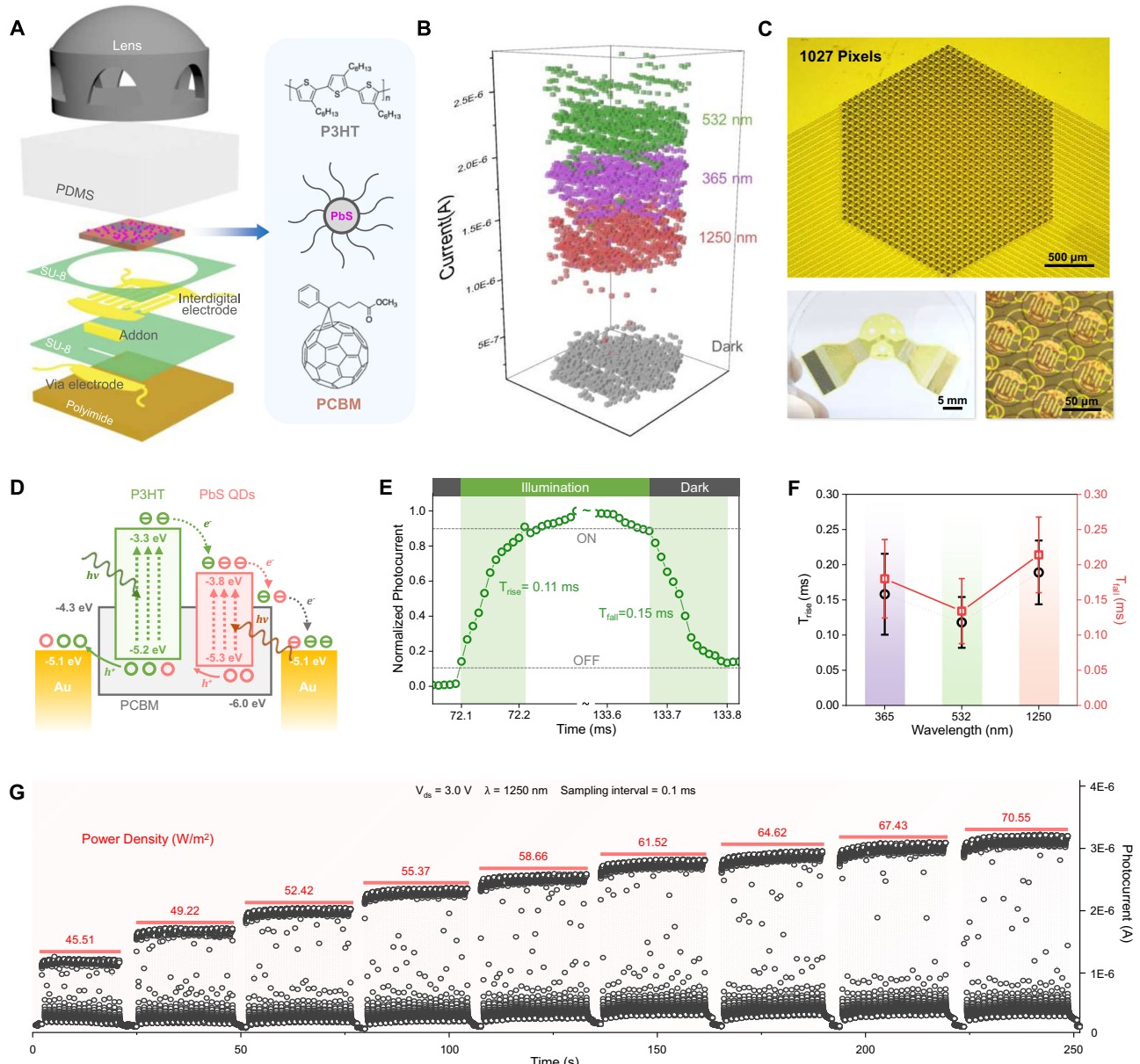

**Fig. 3 | Optoelectrical performance of the P3HT/PCBM/PbS QDs-based photodetector array. A** The vertical disassembly schematic diagram of a bio-CE's ommatidium, The PbS quantum dots doped P3HT/PCBM organic film works as the photosensitive layer of bio-CE; **B** Wide spectrum detection capability test, full frame photoelectric response to ultraviolet-visible-near infrared; **C** Optical images of the 1027 pixels organic photodetector array; **D** The energy band diagram of photo-excited carrier transport in the organic composite films; The tests of bio-CE photodetector pixels reveal 0.1 ms order of photoelectric switching speed (**E**), and the response consistency over a broad spectrum (**F**); Also present outstanding power density sensitivity (**G**).

wavelengths, as quantitatively characterized in Fig. 3B. Optical images of the released device in Fig. 3C reveals the bio-inspired honeycomb pixel arrangement with P3HT/PCBM/PbS QDs-modified active regions. As shown in Supplementary Fig. 10, this configuration achieves a 62.9% fill factor (50 μm pixel diameter, 60 μm inter-pixel spacing), representing an 8.4% improvement over conventional 90° row-column designs (54.5% fill factor). Under the same lens parameters, the filling factor of honeycomb arrangement is significantly higher than that of cross shaped arrangement. When the array area is large, more compound eye channels can be accommodated under the same area, resulting in higher imaging resolution. The innovative 120° cross-arrangement of interdigital electrodes (rows) and wiring connections (columns) extends to 37 peripheral contacts, which interface with the readout circuitry via anisotropic conductive film (ACF) bonding. This

design simultaneously optimizes optical collection efficiency and electrical connectivity. The bio-CE employs a dedicated channel addressing scheme to individually acquire photoelectric responses from each ommatidium. To ensure mechanical durability in curved configurations, we implemented innovative wiring architectures: periodic S-shaped traces in high-strain regions and helical interconnects in transitional zones between the interdigital electrodes and wiring layers. Finite element simulations (Supplementary Fig. 11) demonstrate this hybrid design maintains electrical continuity and interfacial adhesion strength under repeated bending, validating its suitability for conformal integration on curved surfaces.

P3HT/PCBM thin films have been extensively utilized in organic photovoltaics[46,47], photodetectors[48], and flexible electronics[49] owing to their air stability and photoelectric conversion efficiency, we

engineered an advanced organic-inorganic hybrid photodetector by incorporating PbS quantum dots (QDs) into a conventional P3HT/PCBM organic matrix to address the demanding requirements of bio-CE applications. The PbS QDs doping significantly extends the spectral detection range (Supplementary Fig. 12). Absorption spectrum reflects the probability of photon absorption by the material. The high absorption beyond 1000 nm is attributed to the narrow bandgap of PbS QDs, enabling strong infrared (IR) light harvesting. However, not all absorbed photons generate extractable charges due to Shockley-Read-Hall recombination in PbS (high at long wavelengths) and energy loss via thermalization of low-energy photons. Photocurrent depends not only on absorption but also on charge generation efficiency and charge collection efficiency. At 532 nm (visible light), dominated by P3HT/PCBM absorption, which has high charge separation efficiency. Electrons/holes are rapidly transported to electrodes, minimizing recombination. At 1250 nm (IR light), absorption primarily occurs in PbS QDs, surface traps may enhance non-radiative recombination. Low mobility of IR-generated carriers (common in QDs) reduces collection efficiency. Even with 5 times higher absorption, losses in charge extraction suppress the expected photocurrent gain. While PbS enhances IR absorption, its lower charge collection efficiency moderates the photocurrent advantage. This behavior aligns with known QD photovoltaic limitations.

Comprehensive morphological characterization through transmission electron microscopy (TEM) and atomic force microscopy (AFM) (Supplementary Fig. 13) confirms the uniform dispersion of QDs and minimal surface roughness, while the energy band diagram (Fig. 3D) demonstrates optimal energy level alignment for efficient charge transfer, with PbS QDs serving as both spectral sensitizers and charge transport mediators. The energy level alignment in our P3HT/PCBM/PbS QDs system creates an efficient charge collection pathway. PCBM's LUMO level is strategically positioned between the work function of Au electrodes and the LUMO levels of both P3HT and PbS QDs, enabling effective collection of photoexcited electrons across all spectral bands and their subsequent transport to electrodes under applied bias. Meanwhile, the closely matched HOMO levels of P3HT and PbS QDs to the Au work function facilitate barrier-free hole collection, minimizing charge recombination losses. This optimized energy landscape, combined with the cascaded charge transfer pathways demonstrated in previous studies of similar organic systems[50,51], ensures efficient separation and collection of both electrons and holes, contributing to the photodetector's high quantum efficiency and fast response time.

Transient response measurements reveal excellent dynamic performance, with a rise time of 0.11 ms and fall time of 0.15 ms for a typical device (Fig. 3E). This rapid response enables the bio-CE system to achieve a flicker fusion frequency of 1k Hz (Supplementary Fig. 14), surpassing most biological compound eyes (Supplementary Table 1 and 2). As shown in Fig. 3F, the photodetector array demonstrates consistent response speeds of ~0.1 ms across various wavelengths (365 nm, 532 nm, and 1250 nm). Supplementary Fig. 15 further confirms the outstanding wavelength response uniformity across the entire array. The temporal response characteristics under varying illumination intensities (Fig. 3G) reveal that the P3HT/PCBM/PbS QDs hybrid system achieves both ideal metal-semiconductor contact properties and excellent power sensitivity. Supplementary Fig. 16 demonstrates durability, with photo response stability under 10000 operational cycles. Furthermore, the PDMS-encapsulated pixels demonstrate significantly improved environmental stability, representing a longer operational lifetime compared to unencapsulated devices (Supplementary Fig. 17).

## Obstacle avoidance in a comprehensive scenario

The operational principle of our device relies on the detection of contrast and specific spatial patterns. Theoretically, this mechanism should be independent of the light source, as long as sufficient illumination enables the target to reflect light toward the sensor. Consequently, the system is expected to identify objects under natural lighting, since recognition primarily depends on the reflectance difference between the target and its background. We performed quantitative benchmark tests using active illumination and further validated the system's performance with passive targets in natural light. In simulated environments, when the light emitted or reflected by the target object significantly differs from the ambient illumination, the bio-CE device can successfully visualize the scene and be effectively applied to obstacle avoidance and navigation in unmanned vehicles.

Building upon our bionic micro-lens array design and organic photodetector array characterization, we successfully demonstrated bio-inspired obstacle avoidance functionality using an unmanned omnidirectional vehicle (UOV) system equipped with a bio-CE, as illustrated in Fig. 4A. The integrated UOV platform comprises a bio-CE sensor module, custom adapter board, Artix-7 FPGA-based data acquisition-processing board, 3D-printed structural frame, and omnidirectional chassis, achieving real-time spatial perception through synchronized frame acquisition, target localization using coordinate-transformed bio-CE data streams, and adaptive control of all four Mecanum wheels. The system's wiring architecture connects the bio-CE to the customized Artix-7 data acquisition board through an intermediate adapter board with J30J connectors, enabling dynamic target spatial parameter extraction and precise motion control execution, with complete system component specifications and mechanical design details provided in Supplementary Fig. 18.

We employed FL-TPP technology to directly fabricate the micro-lens array onto the organic photodetector array, realizing a bionic apposition compound eye structure with precise one-micro-lens-to-one-pixel alignment, as detailed in Supplementary Note 2 (fabrication process) and Supplementary Fig. 19 (structural characterization). The resulting bio-CE devices were mounted on a custom holder manufactured via low-resolution FL-TPP, ensuring secure integration with a cylindrical surface (radius = 0.6 mm) while maintaining optimal optical spacing (Supplementary Fig. 20), with this optimized integration configuration enabling an ultra-wide 80° (elevation) × 180° (azimuth) field-of-view (Fig. 4B). Leveraging the bio-CE's simplified imaging pattern and the photodetector's high-speed performance, the system attained a 1 kHz frame sampling rate and millisecond-scale motion detection capability, demonstrating exceptional temporal resolution for dynamic tracking applications.

The static target detection and single-frame signal processing capabilities of the bio-CE were evaluated as shown in Fig. 4C. The spatial resolution for distinguishing objects varies with target distance. Therefore, the bio-CE imaging model (Supplementary Fig. 21) and its characterized resolution (Supplementary Fig. 22) served as key references for determining appropriate experimental distances. We positioned objects of varying shapes at distances of 10-25 cm from the bio-CE to systematically assess key performance metrics, including elevation/azimuth angle resolution, multi-target discrimination, distance estimation, contour recognition, and fog-resistant imaging. The results demonstrate that while the bio-CE achieves precise angular parameter extraction, it provides only approximate shape and contour reconstruction due to its bio-inspired ommatidial density and uniform angular distribution in both elevation and azimuth directions. Two further shortcomings were observed. First, when a pattern is imaged by the bio-CE, it undergoes stretching distortion along the axial direction, requiring proportional correction to restore the target's true morphology. Second, the system shows weak or no response to fine shape details such as small holes and thin boundaries, which results from the limited spatial resolution of the cylindrical compound eye channels. Despite these limitations, the bio-CE demonstrates accurate recognition of the target's overall contour and high sensitivity to changes in its position, orientation, and motion state. These functional

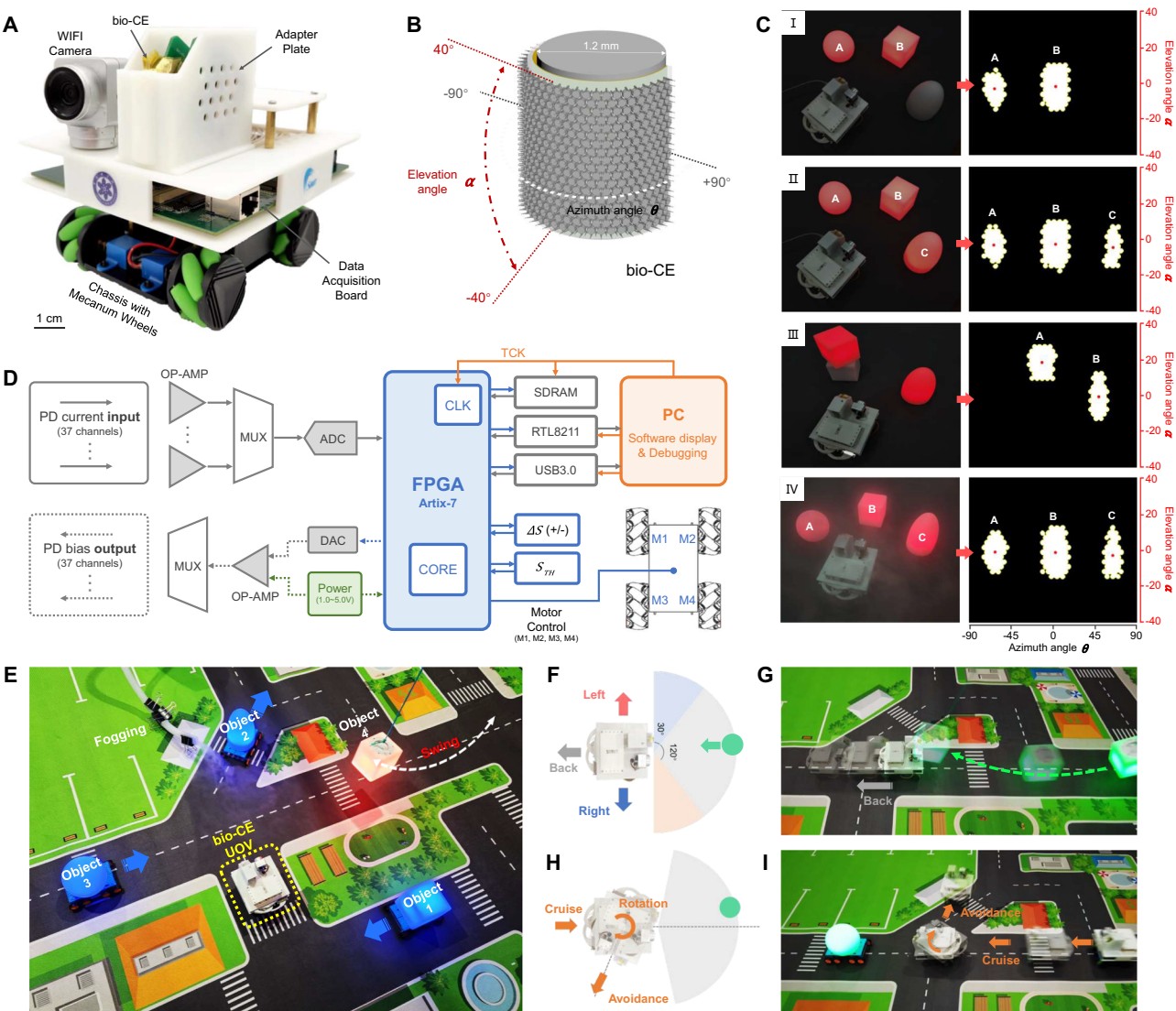

**Fig. 4 | The bio-CE equipped unmanned omnidirectional vehicle (UOV) and obstacle avoidance in a comprehensive scenario. A** Photograph of a UOV and its components; **B** Schematic diagram of the range of the azimuth and elevation angles covered by bio-CE; **C** Influence of different target shape, position, distance, number, and test conditions on bio-CE imaging form; **D** Block diagram of UOV workflow, including bio-CE addressing data acquisition, FPGA real-time frame processing, parameters reading by the host computer, and motion control of the omnidirectional chassis; **E** Image of bio-CE equipped UOV in a comprehensive simulated scenario; The as-prepared UOV can make avoidance or retreat to rapidly approaching targets in different directions (**F**, **G**), or achieve steering and evasion during autonomous cruising (**H**, **I**).

traits align well with findings from studies on arthropod compound eyes, such as those of *Drosophila*. Notably, the integrated bionic setae array effectively prevents droplet condensation through its self-cleaning functionality, maintaining imaging stability comparable to dry conditions even under fogging. Additional static test data, including comprehensive comparisons of angular resolution and shape recognition accuracy, are provided in Supplementary Fig. 23.

Fig. 4D schematically illustrates the complete sensory-processing-actuation pipeline of our bio-CE equipped UOV system. The architecture enables dual functionality: (1) real-time visualization of bio-CE static test results through a host computer interface, and (2) hardware-accelerated implementation of obstacle avoidance algorithms via FPGA programming using Verilog/VHDL. This integrated design facilitates fully autonomous operation with both reactive and proactive avoidance capabilities. We developed two bio-inspired navigation strategies within the 180° azimuth field-of-view to validate system performance (Fig. 4E). In the passive avoidance mode (Fig. 4F, G),

mimicking arthropod escape responses, the system continuously monitors target movements by analyzing activation area changes across five consecutive frames. When targets approaching within the 120° frontal zone demonstrate consistent approach patterns, the UOV executes evasive maneuvers, including backward movement for frontal threats (120° zone) or lateral displacement for side approaches (30° zone). The active avoidance mode (Fig. 4H, I) implements autonomous navigation by maintaining preset cruising speed until detecting obstacles within the FOV, upon which it initiates coordinated braking and steering maneuvers to ensure collision-free traversal. A movie recording of the experiment is available as Supplementary Movie 2, demonstrating the real-time performance of the bio-CE microsystem under dynamic testing conditions. Additionally, results from confirmatory tests in environments with multiple moving obstacles are provided in Supplementary Fig. 24, which systematically characterizes the system's obstacle avoidance accuracy and response dynamics.

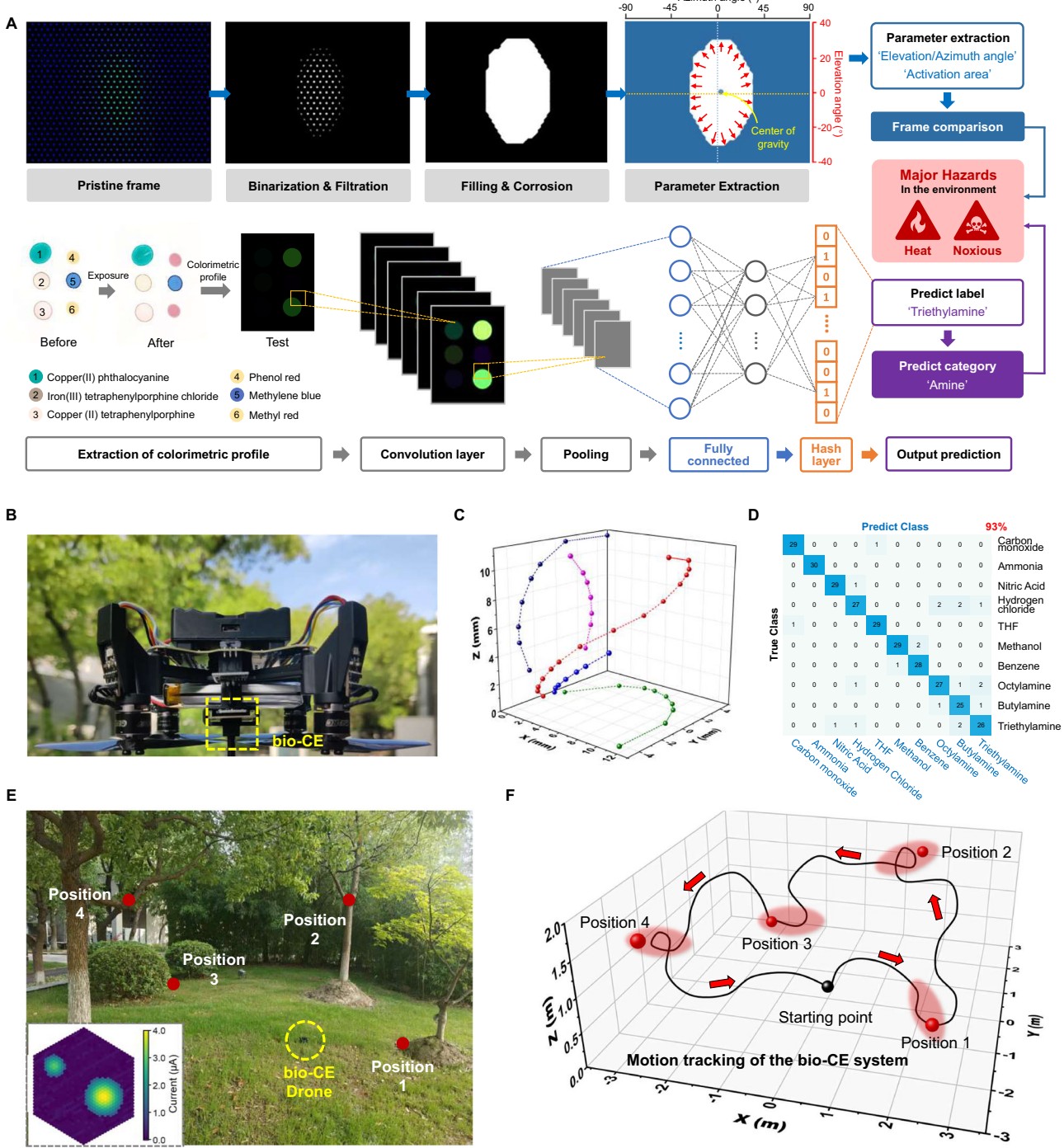

**Fig. 5 | Target recognition and drone motion tracking of the bio-CE system.**
**A** The algorithm process for visual-olfactory fusion in hazardous environments includes visual frame processing and target parameter extraction methods, olfactory convolutional neural networks and hash classification; **B** Photograph of the micro drone equipped with bio-CE; **C** Calculated spatial positions and generated movement path of a moving point light source in 3D space; **D** The recognition accuracy and confusion matrix of 10 hazardous gases using a colorimeter olfactory sensor array; **E** Image of the drone system in a simulated situation, where the drone collects visual-olfactory information from the surrounding environment and autonomously explores target objects. Inset provides the visual imaging result; **F** The exploration trajectory of drone in simulated environments based on visual-olfactory information.

## Target recognition and drone motion tracking

We developed a bionic visual-olfactory fusion algorithm model and validated its target recognition and motion tracking capabilities on a drone equipped with bio-CE. The visual-olfactory fusion perception algorithm, depicted in Fig. 5A, processes visual keyframes through high-contrast enhancement, morphological filling, and erosion to extract precise moving target regions, from which target angle parameters and motion states are derived based on coordinate transformations and activated area dynamics. Simultaneously, a *Drosophila*-inspired locally sensitive hashing (LSH) neural network achieves efficient gas recognition while minimizing parametric complexity. To enhance robustness, the model incorporates scene-dependent feedback, dynamically adjusting the weight ratio between visual and olfactory inputs, thereby ensuring high accuracy in challenging scenarios.

To validate the multimodal fusion performance of visual and olfactory signals, we integrated the bio-CE with a custom-designed circuit board for synchronized signal acquisition, processing, and wireless transmission, thereby constructing a drone-mounted platform (Fig. 5B). Leveraging the system's superior angular selectivity, we deployed it on a commercial drone with open-source flight control to enable real-time tracking of point light sources. By displacing the light source along predefined planar trajectories, we analyzed positional data and reconstructed 3D spatial paths (Fig. 5C), confirming the visual system's target localization capability. Furthermore, the system achieved 93% classification accuracy for toxic volatile compounds in a test dataset comprising 300 samples across 10 categories (Fig. 5D and Supplementary Table 4). The colorimetric sensor array exhibited robust and repeatable responses at Immediately Dangerous to Life or Health (IDLH) concentrations, with signal attenuation correlated to reduced analyte levels (Supplementary Fig. 25), highlighting its utility as a hazardous chemical early-warning platform. Humidity interference tests via a purpose-built valve system confirmed negligible water vapor effects on colorimetric patterns (Supplementary Fig. 26). The simulation scenario for drone-based target recognition and environmental exploration is illustrated in Fig. 5E. The experimental setup consists of spatially distributed point light sources and hazardous chemical emitters, allowing for systematic testing under controlled conditions. By modulating the wavelength of light sources, as well as the type and concentration of gaseous analytes, we simulated a wide range of complex real-world scenarios to rigorously evaluate system performance. Thus, the drone can be autonomously controlled to explore the environment while simultaneously recording trajectories, with real-time visual and olfactory data displayed on an integrated monitoring interface (Fig. 5F).

## Discussion

We report an insect-scale cylindrical bionic apposition compound eye with wide-angle FOV, broad-spectrum detection and rapid gases response capability. The FL-TPP method effectively overcomes the lens-pixel mismatch in previous artificial compound eye studies and bypasses the traditional curved micro-lens array-CMOS compound eye construction route. Additionally, the unique optical design provides bio-CE with excellent interocular optical isolation. The 0.1 ms photoelectric response realizes exceptionally high sensitivity for wide-angle moving target detection and proximity avoidance. Moreover, the coordination chemistry of metal complexes and the proton response of pH indicators work together to achieve rapid and visual detection of complex gas environments.

The target detection capability of bio-CE system is related to the number and integration of compound eye array. Under the premise of the data transmission volume upper limit, it can be considered to further increase the 1 kHz sampling frequency by turning on/off certain data acquisition channels and/or merging adjacent pixels into macro pixels to approach the photoelectric response limit. Combined with single-dimensional imaging mode, it can effectively extract target spatial parameters, making the bio-CE system an excellent candidate for unmanned platforms with limited computing resources and power supply, especially suitable for panoramic motion detection and obstacle avoidance in continuously changing environments. Further improvements include efforts to reduce the size and power consumption of microsystems, refine the transmission and fusion algorithm of visual-olfactory information, and expand the application scope in micro-unmanned intelligent agents.

## Methods

### Fabrication of the cylindrical bionic compound eye (bio-CE)
A 1000 nm Aluminum layer is deposited by sputter coater on the Si substrate as the sacrifice layer. Next, a 10 μm polyimide film is spin-coated and annealed onto the sacrifice layer as the flexible base of the double-layer wiring organic photodetector (OPD) array. Bending wiring and interdigital electrodes (10/100 nm, Cr/Au) are patterned by photolithography and electron beam evaporation, and a patterned SU8 layer isolates each wiring layer. After the release window is etched using oxygen plasma, the OPD array is released from the Si substrate in diluted hydrofluoric acid. P3HT/PCBM/PbS quantum dots (QDs) solution is spin-coated onto the OPD array in a glove box ($N_2$ atmosphere) and anneals at 120 °C for 45 min, after which the OPD array is encapsulated using a pre-swelled PDMS film on an 80 °C hot plate. Femtosecond laser two-photon polymerization (FL-TPP) method is applied on the upper side of the packaging PDMS film to fabricate the microlens array (MLA) with polymer resin IP-DIP (Nanoscribe). The femtosecond laser's wavelength is 780 nm, and the pulse duration is 90 fs. MLA is achieved using a designed aligned printing process on GT2 Professional II (Nanoscribe) by measuring and adjusting the relative position and angle between the micro-lens model and the OPD array. MLA development is applied in PGMEA (15 min) and IPA (5 min), followed by a naturally air-dried process in a nitrogen environment.

The millimeter scale cylindrical fixture is fabricated by a standard 3D Large-feature (LF) printing set using resin IP-Q (Nanoscribe) based on a silicon substrate. In order to separate the fixture from the substrate under thermal stress, we have designed several conical round table structures at the bottom of the fixture. The complete bio-CE is obtained by adequately clipping the OPD-MLA device and mounting it in this cylindrical fixture.

### Design and characterization of the bionic micro-lens array
The design of the focal length, spot size, effective acceptance angle, and other parameters of the micro-lens is realized by commercial software ZEMAX (ANSYS Co.). The lens-holder and lens-setae structure models are implemented using SOLIDWORKS. The interocular bionic setae structure is arranged in the crevices of the MLA, its height is similar to the diameter of the micro-lens and will not affect the imaging performance within the receiving angle of the lens. We use several masks of different shapes on the Olympus CX43 microscope platform to verify the imaging performance of the MLA. Commercial CMOS sensors (OmniVision Tech.) are utilized to test the optical isolation between ommatidia. The scanning electron microscope images of a curved MLA are acquired by Hitachi S4800. The surface topography data of a single micro-lens is measured by the Dektak XTL stylus profilometer.

### Characterization and measurement of the organic photodetector array
20 mg of [6,6]-Phenyl C61 butyric acid methyl ester (PCBM, 160848-22-6, Aladdin), 20 mg Poly(3-hexylthiophene-2,5-diyl) (P3HT, 104934-50-1, Aladdin) and 5 mg PbS quantum dot (Aladdin) are dissolved in 1 mL of anhydrous chlorobenzene (Aladdin) in order, and let stirring overnight (20 h) at 70 °C in a glove box under nitrogen atmosphere. The configured organic solution is then spin-coated onto the interdigital electrode array at 2000 rpm. The atomic force microscopy (AFM) characterization of P3HT/PCBM/PbS QDs film is performed by Bruker MultiMode 8-HR, and the transmission electron microscope (TEM) characterization of PbS quantum dots is applied on a Titan G2 platform. The single-channel device photoelectric response test is performed by the Keithley 2700 and a manual probe station. The light sources used in the test include adjustable power lasers of various wavelengths and corresponding laser beam expanders. An optical power meter (OceanOptics Co.) is used to calibrate the optical power density of different wavelength light sources.

### Construction and measurement of the olfactory colorimetric array
The colorimetric sensor array was fabricated using a carefully optimized ink formulation containing six distinct chemo-responsive

indicators: copper tetraphenyl porphyrin (CuTPP) and iron tetraphenyl porphyrin chloride (FeTPPCl) as metalloporphyrin-based sensors, copper phthalocyanine (CuPc) as a coordination complex, along with three pH-sensitive dyes - methyl red, bromophenol blue, and phenol red. This selective combination of colorimetric indicators provides complementary molecular recognition properties for enhanced discrimination of volatile analytes through distinct chromatic fingerprint patterns. The sensing solution was prepared by dissolving 0.5–1 wt% of each color developer in an ethanol/water (3:1 v/v) mixture with 1% polyvinylpyrrolidone (PVP) as adhesive matrix. The array was fabricated using precision inkjet printing technology with optimized waveform control to achieve uniform droplet deposition at 1 mm inter-spot spacing, ensuring minimal cross-contamination while maintaining excellent spot-to-spot reproducibility. Color development typically occurred within 30–120 seconds of analyte exposure, after which high-resolution array images were acquired using a flatbed scanner. Subsequent image processing involved spot localization, RGB color space analysis, and multivariate pattern recognition to extract quantitative olfactory information.

### Data acquisition system design

The OPD array is connected to the PCB adaptor by the ACF bonding process, then connected to the data acquisition board via customized 37 channels J30J connector cables (HAO-AIXI Electronics). We use the ARTIX-7 FPGA from Xilinx Co. as the microprocessor of the data acquisition peripheral circuit, which control the shift register through the clock signal to perform voltage operations on the data lines of the OPD array. The power supply and motor control of the four Mecanum wheels are also realized by the corresponding interfaces of the peripheral circuits. The automatic/passive obstacle avoidance strategy and frame process methods are first edited and verified in MATLAB, then transferred into the Verilog Hardware Description Language (Verilog HDL) and compiled in FPGA.

### Data processing of olfactory colorimetric array on drone

To ensure reliable characterization of gas distribution at each sampling location, the drone was programmed to maintain a stable hover for a duration of 60 sec. This predetermined period allows sufficient time for the olfactory colorimetric sensor array to undergo full adsorption and chemical reaction with the target analytes, thereby ensuring measurable and stable color transitions. Throughout the flight mission, the onboard data acquisition system synchronously records GPS coordinates, high-precision timestamps, and raw multi-channel sensor readings at each of the five predefined positions. All data are stored locally on the drone's storage module and retrieved post-landing for subsequent analysis. The collected raw RGB data from the colorimetric array at each location are processed using a lightweight machine learning model previously trained on laboratory-calibrated exposure experiments. This model translates the complex color shift patterns into quantitative gas concentration values and identifies the presence of specific volatile compounds. The analyzed gas information is then accurately mapped back to its corresponding geographic position using the synchronized GPS and temporal metadata.

While real-time gas detection remains challenging due to the inherent latency of colorimetric reaction and analysis, the implemented post-mission processing pipeline robustly compensates for this delay and ensures reliable spatial mapping of gas information. For applications requiring immediate feedback, future system iterations could incorporate onboard image processing capabilities, edge-computing optimized algorithms, or a hybrid sensor architecture combining colorimetric arrays with complementary real-time electrochemical or MOS sensors.

### Data availability

All relevant data in this study are available from the corresponding author upon request.

### Code availability

The custom MATLAB scripts used for spatial object detection based on cylindrical compound eye structure are available from Zenodo (https://doi.org/10.5281/zenodo.17773754).

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

## Acknowledgements

This work was supported by National Science and Technology Major Project from the Minister of Science and Technology of China (grant no. 2018AAA0103100, T.H.T.), National Natural Science Foundation of China (grant no. 62236005, N.Q.), National Science Fund for Excellent Young Scholars (grant no. 61822406, N.Q.), Key Research Program of Frontier Sciences, CAS (grant no. ZDBS-LY-JSC024, N.Q.) and Fund of Youth Innovation Promotion Association CAS (grant no. 2022234, N.Q.).

## Author contributions

J.W., S.W. and N.Q. were responsible for instrument design, production, experiment and data analysis. J.W., S.W. and T.H.T. wrote the paper. T.H.T. is responsible for the guidance of the whole process. All authors were involved in revising the manuscript.

## Competing interests

The authors declare no competing interests.
