## [Transparent Peer Review file · Nature Communications]

An insect-scale artificial visual-olfactory bionic compound eye

Corresponding Author: Professor Tiger Tao

Version 0:

Reviewer comments:

Reviewer #1

(Remarks to the Author)

The authors have reported an integrated cognitive system that incorporates a bio-inspired vision module, composed of two-photon polymerization-based microlens arrays and flexible photodetectors, as well as an inkjet printing-based olfactory sensor array. The workload is quite significant and the device is well-constructed, but the experimental imaging results are somewhat disappointing compared to the complexly manufactured device. In addition, the logical rationale for integrating the colorimetric olfactory sensor array with insect-inspired systems appears to be somewhat insufficient. The manuscript needs an extensive revision of the following suggestions.

1. In Figure 4C, it is notably challenging to infer the actual shape of the target solely from the acquired image. Furthermore, significant distortion appears to be induced by the object's position. Therefore, a detailed characterization of the distortion information within these images seems crucial for proper interpretation.
2. All targets used in imaging experiments in this paper include light sources. Natural insects can recognize targets reflected in natural light. I wonder if the fabricated device can identify objects in natural light, and if not, additional discussion is required.
3. Imaging targets used in this paper were located in 10-25 cm. The resolution for distinguishing objects changes depending on the distance to the target, so it would be recommended to include a theoretical graph of the resolution.
4. The color metric sensor mounted on a drone is considered difficult to detect in real time during actual operation at a distance. Please specify a method to identify the gas information at each location when the drone returns after passing through a total of 5 positions.
5. In Supplementary Figure 2, more specific descriptions about the lens-photodetector alignment fabrication process on each figure are required.
6. In Supplementary Figure 4, it appears as if two colors are mixed within a single pixel. What causes this phenomenon?
7. In Supplementary Figure 9, the absorption spectra value for P3HT/PCBM/PbS above approximately 1000 nm is about 20%, which is approximately 5 times higher than that at about 500 nm. However, in Figure 3B, the current at 532 nm shows only about a 1.6-fold difference compared to 1250 nm. A specific explanation for this discrepancy is required.
8. If the authors have a video of the experiment in Figure 4E~I, please submit it as a Supplementary video.

Reviewer #2

(Remarks to the Author)

This manuscript presents a breakthrough in bio-inspired vision systems by developing an insect-scale cylindrical bionic apposition compound eye with multifunctional capabilities. The bio-CE enables wide field-of-view imaging, natural interocular isolation, a 1 kHz flicker fusion frequency and color response to various hazardous chemicals, resulting in high

sensitivity to moving objects and rapid response to environmental gases. To help readers appreciate the advance more fully, there several suggestions need to be modified.

1. The authors highlighted that the bio-inspired compound eye (bio-CE) employs a 'single-pixel-per-lens' architecture to achieve exceptional motion sensitivity. However, compared to conventional CMOS-based designs, could this sparse data acquisition approach compromise spatial resolution? Please provide a quantitative comparison of the trade-offs between spatial resolution and motion detection sensitivity for both architectures, along with specific experimental data supporting the superiority of the proposed design over traditional solutions.
2. Regarding the individual microlenses with narrow acceptance angles that can inspire isolation effect in the optoelectronic device. Has optical crosstalk been quantitatively measured to verify the effectiveness of this isolation? Specifically, what percentage of crosstalk suppression was achieved compared to conventional designs? Please provide supplemental experimental data.
3. While the reported 62.9% fill factor represents an improvement over conventional designs, the comparison to "90° row-column designs (54.5%)" lacks context. Please provide: A reference for the baseline 54.5% fill factor. An explanation of how the 120° cross-arrangement directly enables this improvement.
4. The experiments appear conducted under controlled conditions. How does the system perform in: Dynamic environments with multiple moving obstacles? Low-contrast or variable lighting condition?

Version 1:

Reviewer comments:

Reviewer #1

(Remarks to the Author)

While it is somewhat regrettable that most responses primarily acknowledged the current system's limitations rather than presenting groundbreaking solutions, we nevertheless confirmed the significance of this research through the appended Supplementary Video. However, we found the discussion regarding the MTF (Supplementary Figure 22) to be inappropriate. Specifically, the presented graph represents the image results of a fabricated microlens array without the integration of sensors, rather than the resolution of the visual system. Given that each microlens is associated with a single pixel of 50 μm , resulting in a Nyquist frequency of 10 lp/mm, the overall system cannot practically achieve the MTF performance indicated. Therefore, the terminology needs to be corrected from "Resolution of the visual system" to "Resolution of the fabricated microlens array".

Reviewer #2

(Remarks to the Author)

The authors have addressed all questions and it can be published at its current form.

made.

REVIEWER COMMENTS

Reviewer #1 (Remarks to the Author):

The authors have reported an integrated cognitive system that incorporates a bio-inspired vision module, composed of two-photon polymerization-based microlens arrays and flexible photodetectors, as well as an inkjet printing-based olfactory sensor array. The workload is quite significant and the device is well-constructed, but the experimental imaging results are somewhat disappointing compared to the complexly manufactured device. In addition, the logical rationale for integrating the colorimetric olfactory sensor array with insect-inspired systems appears to be somewhat insufficient. The manuscript needs an extensive revision of the following suggestions.

1. In Figure 4C, it is notably challenging to infer the actual shape of the target solely from the acquired image. Furthermore, significant distortion appears to be induced by the object's position. Therefore, a detailed characterization of the distortion information within these images seems crucial for proper interpretation.

We sincerely appreciate the reviewer's insightful observation regarding Figure 4C. We acknowledge that the image alone may not fully convey the target's true geometry due to distortion effects. We designed multiple static scenarios to verify the target detection performance of the biomimetic compound eye microsystem, including multi-target detection, distance detection, direction angle testing, and performance testing under foggy conditions. Figure 4C (I) shows the multi-target testing conducted in an ideal contrast environment, where the unlit light source cannot provide sufficient photoelectric response and is considered as interference noise during the filtering process. The luminous light sources A and B differ significantly in the horizontal angle direction. However, the hemispherical shape (A) and cubic shape (B) have a certain degree of shape distortion due to the presence of cylindrical structures; Figure 4C (II) and (III) show the imaging results of the biomimetic compound eye microsystem on the target light source at different distances and elevation angles. The detection shows that, without considering the target shape and size, it is theoretically possible to cover a 180° horizontal angle and a 90° elevation angle. Thanks to the design of biomimetic bristles, the biomimetic compound eye can still maintain a relatively stable imaging ability under artificial fog conditions and can accurately image targets in different directions. However, due to the scattering effect of the light source in the fog environment, the detection results in Figure 4C (IV) slightly expand the activation area compared to the no fog condition and have a certain ability to distinguish the morphology of the target.

We have also acknowledged this as a limitation of our imaging approach in the Results section (Page 13).

“It should be noted that the images obtained in this study are subject to position-dependent geometric distortion, which may complicate the direct interpretation of object morphology. Two further shortcomings were observed. First, when a pattern is imaged by the bio-CE, it undergoes stretching distortion along the axial direction, requiring proportional correction to restore the target’s true morphology. Second, the system shows weak or no response to fine shape details such as small holes and thin boundaries, which results from the limited spatial resolution of the cylindrical compound eye channels. Despite these limitations, the bio-CE demonstrates accurate recognition of the target’s overall contour and high sensitivity to changes in its position, orientation, and motion state. These functional traits align well with findings from studies on arthropod compound eyes, such as those of *Drosophila*. While we have characterized this effect (Supplementary Figure 23), future work could employ real-time distortion correction algorithms for improved accuracy.”

2. All targets used in imaging experiments in this paper include light sources. Natural insects can recognize targets reflected in natural light. I wonder if the fabricated device can identify objects in natural light, and if not, additional discussion is required.

We thank the reviewer for raising this important question regarding the operational applicability of our device under natural lighting conditions. In response, we have added a detailed discussion and referenced it accordingly in the Results section (Page 11).

“The operational principle of our device relies on the detection of contrast and specific spatial patterns. Theoretically, this mechanism should be independent of the light source, as long as sufficient illumination enables the target to reflect light toward the sensor. Consequently, the system is expected to identify objects under natural lighting, since recognition primarily

depends on the reflectance difference between the target and its background. We performed quantitative benchmark tests using active illumination and further validated the system's performance with passive targets in natural light. In simulated environments, when the light emitted or reflected by the target object significantly differs from the ambient illumination, the bio-CE device can successfully visualize the scene and be effectively applied to obstacle avoidance and navigation in unmanned vehicles.”

We have supplemented the specific obstacle avoidance of the biomimetic compound eye system in natural light environments (Supplementary Figure 27).

3. Imaging targets used in this paper were located in 10-25 cm. The resolution for distinguishing objects changes depending on the distance to the target, so it would be recommended to include a theoretical graph of the resolution.

We appreciate the reviewer’s suggestion to clarify the relationship between resolution and target distance. Within a distance of 10-25 cm, due to the light gathering and eye isolation of the micro lens, when the output current exceeds the set threshold, the pixel can be considered activated, thereby achieving imaging.

We establish the following bio-CE imaging model based on the illumination angle of the light source and the target distance (Supplementary Figure 21), where S is the distance between the microlenses.

$$L1 = 2 \tan\left(\frac{\theta_0}{2}\right) \{r - (R+RL)\}$$

$$L2 = \left\{ 2(R+RL) - \frac{2(R+RL)^2}{r} \right\} \tan\left(\frac{\theta_0}{2}\right)$$

$$n = \left[\frac{\pi \tan^2\left(\frac{\theta_0}{2}\right) (R+RL)}{\frac{\sqrt{3}}{2} S^2} \times \frac{\{r - (R+RL)\}^2}{r} \right]$$

MTF represents the resolution of the visual system. Observe a specific target card using a lens and read the MTF value, with 0.5 being the resolution contrast optimization point. The maximum resolution obtained from simulation is 416 lp/mm (2.4 μm), while the measured resolution is 3.7 μm, which is better than the 70 μm of the human eye.

The Supplementary Figure 22 is shown below.

We have also acknowledged this as a limitation of our imaging approach in the Results section (Page 12).

“The spatial resolution for distinguishing objects varies with target distance. Therefore, the bio-CE imaging model (Supplementary Figure 21) and its characterized resolution (Supplementary Figure 22) served as key references for determining appropriate experimental distances.”

4. The color metric sensor mounted on a drone is considered difficult to detect in real time during actual operation at a distance. Please specify a method to identify the gas information at each location when the drone returns after passing through a total of 5 positions.

We acknowledge the reviewer's valid point regarding the challenges of real-time gas detection with colorimetric sensors under dynamic drone operation. We have now explicitly addressed this limitation of colorimetric sensing in the Methods section (Page 23).

“To ensure reliable characterization of gas distribution at each sampling location, the drone was programmed to maintain a stable hover for a duration of 60 seconds. This predetermined period allows sufficient time for the olfactory colorimetric sensor array to undergo full adsorption and chemical reaction with the target analytes, thereby ensuring measurable and stable color transitions. Throughout the flight mission, the onboard data acquisition system synchronously records GPS coordinates, high-precision timestamps, and raw multi-channel sensor readings at each of the five predefined positions. All data are stored locally on the drone's storage module and retrieved post-landing for subsequent analysis. The collected raw RGB data from the colorimetric array at each location are processed using a lightweight machine learning model previously trained on laboratory-calibrated exposure experiments. This model translates the complex color shift patterns into quantitative gas concentration values and identifies the presence of specific volatile compounds. The analyzed gas information is then accurately mapped back to its corresponding geographic position using the synchronized GPS and temporal metadata.

While real-time gas detection remains challenging due to the inherent latency of colorimetric

reaction and analysis, the implemented post-mission processing pipeline robustly compensates for this delay and ensures reliable spatial mapping of gas information. For applications requiring immediate feedback, future system iterations could incorporate onboard image processing capabilities, edge-computing optimized algorithms, or a hybrid sensor architecture combining colorimetric arrays with complementary real-time electrochemical or MOS sensors.”

5. In Supplementary Figure 2, more specific descriptions about the lens-photodetector alignment fabrication process on each figure are required.

We thank the reviewer for the valuable suggestion to enhance the fabrication details. In response, we have substantially expanded the schematic descriptions in Supplementary Figure 2 and revised Supplementary Note 2 to include a comprehensive, step-by-step description of the fabrication process.

Supplementary Note 2. Lens-photodetector alignment fabrication process.

In this supplementary note, we conducted a detailed explanation of the alignment manufacturing scheme between the micro-lens array and the photoelectric detector array, and used schematic diagrams (Supplementary Figure 2) to display the design details in each step.

Correction of coordinate systems:

Accurate alignment of the multi-level coordinate systems is essential for ensuring precise micro-scale fabrication. Prior to structural printing, we performed a comprehensive calibration procedure to synchronize the galvanometer, stage, and writing area coordinate systems according to the manufacturer's established protocol. The calibration process commenced with the printing of a fine cross-marker on a prepared silicon substrate (Supplementary Figure 2A), whose geometric center was designated as the origin of the galvanometer coordinate system. This marker serves as a critical reference point for all subsequent alignment steps.

The calibration protocol continued with the adjustment of the laser direct-writing area's reference frame. Using the software interface, we dynamically manipulated the reference grid by dragging and zooming operations until perfect alignment was achieved with the pre-printed cross-marker (Supplementary Figure 2B). This crucial step established spatial correspondence between the virtual writing coordinates and physical substrate positions, effectively registering the origin within the writing zone. The alignment precision was verified through stage movements and real-time image processing, ensuring coordinate consistency across all systems.

To validate the calibration accuracy, we employed a functional verification approach by printing a single microlens structure at the defined coordinate origin (Supplementary Figure 2C). Subsequent microscopic inspection confirmed the lens was precisely positioned relative to the

alignment marker, with a measured placement error of less than 1 μm . This quantitative validation protocol not only verified the coordinate system alignment but also confirmed the system's readiness for complex multi-structure fabrication. The entire calibration process, including verification, required approximately 15 minutes and was performed once per substrate loading to maintain spatial accuracy throughout the fabrication session.

Loading micro-lens array model and photodetector array installation:

Following the establishment of the coordinate origin, the STL model of the micro-lens array was imported into the fabrication software for printing file generation. To ensure precise alignment with the photodetector array on the substrate, the initial position of the lens model was designed with a crossbar marker. The complete micro-lens array was fabricated sequentially in a snake-like patterning scheme (Supplementary Figure 2D), optimizing the writing path for efficiency and minimal stage movement.

The encapsulated photodetector array was then mounted on the printing stage, allowing clear observation of alignment markers and pixel structures within the direct-writing area (Supplementary Figure 2E). To achieve accurate registration between the lens model and underlying pixels, angular displacement correction was performed and the starting position was finalized prior to initiating the final printing process. This alignment verification ensured precise spatial correspondence between the fabricated optical elements and the photodetector pixels.

Rotating correction and alignment printing of the lens model:

The mounting process of the photodetector array on the substrate holder introduces inherent angular misalignment due to mechanical tolerances in the fixture system. This randomness prevents guaranteed orientation consistency between the pixel arrangement and the micro-lens model's printing direction, potentially causing significant optical performance degradation. To address this critical alignment challenge, we implemented a multi-point registration protocol for precise rotational correction of the micro-lens model.

The angular deviation was quantified using a set of alignment markers pre-patterned at four corners of the photodetector array (Supplementary Figures 2F, G and H). These markers form a reference coordinate system that captures the actual orientation of the detector pixels relative to the printing coordinate system. Through coordinate transformation processing (Supplementary Figure 2I), which involves calculating the centroid positions of all markers and performing singular value decomposition, the precise rotational offset was determined and automatically applied to the micro-lens model.

Following model transformation, the updated micro-lens array was re-imported into the writing software (Supplementary Figure 2J and K). The verification phase confirmed successful

registration, with residual alignment errors measuring less than 1 μm across the entire printing area. This sub-pixel level accuracy ensures precise spatial correspondence between each micro-lens optical axis and the underlying photodetector pixels, which is essential for maintaining optimal light collection efficiency and cross-talk minimization in the final bio-CE device.

Micro lens array splicing printing:

The micro-lens array was fabricated using a group-wise splicing strategy, wherein 20 lenses (arranged in a 4×5 configuration) were aligned and printed as a single unit. This process was repeated systematically until complete coverage of all photodetector pixels was achieved. Compared to single-exposure full-array printing, the splicing approach offers significantly improved alignment accuracy by minimizing error accumulation caused by piezoelectric stage drift over large displacements.

Furthermore, the group-wise method enables real-time process monitoring and intervention. Any processing anomalies—such as bubble formation or localized overexposure—can be immediately detected and addressed without compromising the entire array. This segmented fabrication strategy thus enhances yield and ensures consistent optical quality across the bio-inspired compound eye device.

6. In Supplementary Figure 4, it appears as if two colors are mixed within a single pixel. What causes this phenomenon?

We appreciate the reviewer for this insightful observation. The apparent pixel misalignment and color variations arise from interpolation artifacts introduced during image post-processing, and do not reflect the actual sensor output. It is important to note that each pixel corresponds to a single-color value, which represents its normalized photoresponse. We have updated Supplementary Figure 4 accordingly to better illustrate this point, and the revised version is provided below.

7. In Supplementary Figure 9, the absorption spectra value for P3HT/PCBM/PbS above approximately 1000 nm is about 20%, which is approximately 5 times higher than that at about 500 nm. However, in Figure 3B, the current at 532 nm shows only about a 1.6-fold difference compared to 1250 nm. A specific explanation for this discrepancy is required.

We thank the reviewer for this careful comparison of the datasets. The apparent discrepancy stems from fundamental differences between absorption characteristics and photocurrent generation mechanisms. To address this point, we have added detailed explanatory notes and referenced this discussion in the Results section (Page 10).

“Absorption spectrum reflects the probability of photon absorption by the material. The high absorption beyond 1000 nm is attributed to the narrow bandgap of PbS QDs, enabling strong infrared (IR) light harvesting. However, not all absorbed photons generate extractable charges due to Shockley-Read-Hall recombination in PbS (high at long wavelengths) and energy loss via thermalization of low-energy photons. Photocurrent depends not only on absorption but also on charge generation efficiency and charge collection efficiency. At 532 nm (visible light), dominated by P3HT/PCBM absorption, which has high charge separation efficiency. Electrons/holes are rapidly transported to electrodes, minimizing recombination. At 1250 nm (IR light), absorption primarily occurs in PbS QDs, surface traps may enhance non-radiative recombination. Low mobility of IR-generated carriers (common in QDs) reduces collection efficiency. Even with 5 times higher absorption, losses in charge extraction suppress the expected photocurrent gain. While PbS enhances IR absorption, its lower charge collection efficiency moderates the photocurrent advantage. This behavior aligns with known QD

photovoltaic limitations.”

8. If the authors have a video of the experiment in Figure 4E~I, please submit it as a Supplementary video.

We thank the reviewer for this constructive suggestion. As recommended, we have now included a high-resolution video as **Supplementary Video 2** to visually demonstrate its application in unmanned vehicle obstacle avoidance. This video is now referenced in the Results section (Page 14).

Reviewer #2 (Remarks to the Author):

This manuscript presents a breakthrough in bio-inspired vision systems by developing an insect-scale cylindrical bionic apposition compound eye with multifunctional capabilities. The bio-CE enables wide field-of-view imaging, natural interocular isolation, a 1 kHz flicker fusion frequency and color response to various hazardous chemicals, resulting in high sensitivity to moving objects and rapid response to environmental gases. To help readers appreciate the advance more fully, there several suggestions need to be modified.

1. The authors highlighted that the bio-inspired compound eye (bio-CE) employs a 'single-pixel-per-lens' architecture to achieve exceptional motion sensitivity. However, compared to conventional CMOS-based designs, could this sparse data acquisition approach compromise spatial resolution? Please provide a quantitative comparison of the trade-offs between spatial resolution and motion detection sensitivity for both architectures, along with specific experimental data supporting the superiority of the proposed design over traditional solutions.

We appreciate the reviewer's insightful comment regarding the potential trade-off between spatial resolution and motion detection sensitivity in our bio-inspired compound eye (bio-CE) design. The reviewer rightly points out that the 'single-pixel-per-lens' architecture may limit spatial resolution compared to conventional CMOS sensors.

We have provided detailed supplementary explanations in the Results section (Page 6).

We fully agree that the spatial resolution of our bio-CE is lower than high-density CMOS sensors. This is an inherent trade-off of the sparse sampling architecture, which prioritizes wide-field motion detection over static image detail. However, our bio-CE achieves >1000 Hz effective motion sampling. This aligns with bio-inspired vision goals where speed and energy efficiency are critical (e.g., drone navigation, robotic collision avoidance). The resolution-sensitivity trade-off is intentional for target applications. Inspired by the remarkable obstacle avoidance capabilities of *Drosophila*, we present a bio-inspired compound eye activation model (Supplementary Note 3) that achieves high-sensitivity motion detection through low-resolution image processing. This biomimetic approach demonstrates how efficient motion perception can be realized with minimal computational resources, mirroring the energy-efficient visual processing observed in insect vision systems. In summary, while the bio-CE's spatial resolution is lower than CMOS, its motion detection capabilities (speed, sensitivity, power efficiency) are superior for dynamic vision tasks.

2. Regarding the individual microlenses with narrow acceptance angles that can inspire

isolation effect in the optoelectronic device. Has optical crosstalk been quantitatively measured to verify the effectiveness of this isolation? Specifically, what percentage of crosstalk suppression was achieved compared to conventional designs? Please provide supplemental experimental data.

We appreciate the reviewer's insightful question regarding optical crosstalk in our bio-inspired compound eye (bio-CE) design. The narrow acceptance angles of individual microlenses are indeed critical for isolation, and we have quantitatively verified crosstalk suppression through both optical simulations and experimental measurements.

We have provided detailed supplementary explanations in the Results section (Page 9).

“When the incidence angle exceeds 30°, the spot energy is rapidly attenuated, indicating the optical isolation effect of the short focal length micro-lens array design, as detailed in Supplementary Figure 7 and Figure 8.”

ASF characterizes the lens receiving angle and the cutting ability of edge rays. The measured receiving angle is flat, with good convergence and stable energy. After leaving the receiving angle, the energy rapidly decreases without scattering peaks. The full width at half maximum (FWHM) after Gaussian fitting represents the size of the receiving angle, which is determined by the combination of lens NA and detector area.

The Supplementary Figure 7 is shown below.

It shows in detail a certain situation on the imaging surface of the light spot during the gradual change of the incident light angle of a single lens. As the incident angle gradually increases, the trailing phenomenon formed by the secondary light spot is obvious, but its intensity is much smaller than that of the central main light spot; The equivalent photoelectric detection surface size of a flexible photodetector is about $20 \times 20 \mu\text{m}$. Therefore, when the incident angle is

greater than 6° , the light spot will detach from the photosensitive surface and cannot cause pixel photoelectric response.

The Supplementary Figure 8 is shown below.

3. While the reported 62.9% fill factor represents an improvement over conventional designs, the comparison to "90° row-column designs (54.5%)" lacks context. Please provide: A reference for the baseline 54.5% fill factor. An explanation of how the 120° cross-arrangement directly enables this improvement.

We appreciate the opportunity to clarify these important technical points. Fill factor is a key parameter that characterizes the compactness and array density of micro lenses. Below we provide both the requested reference and a detailed geometrical analysis.

The Supplementary Figure 10 is shown below.

The filling factor of a 120° cross-arrangement circular lens array can be calculated as follows.

$$FF_{\text{hexagon}} = \frac{2\pi R_{\text{lens}}^2}{\sqrt{3} S^2}$$

The filling factor of a 90° row-column circular lens array can be calculated as follows.

$$FF_{cubic} = \frac{\pi R_{lens}^2}{S^2}$$

$$\frac{R_{lens}}{S} = \frac{5}{12}$$

We have provided detailed supplementary explanations in the Results section (Page 10).

“Under the same lens parameters, the filling factor of honeycomb arrangement is significantly higher than that of cross shaped arrangement. When the array area is large, more compound eye channels can be accommodated under the same area, resulting in higher imaging resolution.”

4. The experiments appear conducted under controlled conditions. How does the system perform in: Dynamic environments with multiple moving obstacles? Low-contrast or variable lighting condition?

We thank the reviewer for underscoring these essential practical considerations. In response, we have incorporated a detailed analysis and referenced it in the Results section (Page 14) to address these points.

“A video recording of the experiment is available as **Supplementary Video 2**, demonstrating the real-time performance of the bio-CE microsystem under dynamic testing conditions. Additionally, results from confirmatory tests in environments with multiple moving obstacles are provided in **Supplementary Figure 24**, which systematically characterizes the system's obstacle avoidance accuracy and response dynamics.”

To validate the obstacle avoidance capability of the bio-CE microsystem, we constructed an integrated experimental scenario incorporating multiple complex avoidance processes (Supplementary Figure 24). The scenario design incorporates both stationary and dynamic obstacles and includes challenging conditions such as inclined surfaces and fog interference. Under these configurations, the active and passive obstacle avoidance performance of the bio-CE microsystem was systematically evaluated and analyzed.

In the passive obstacle avoidance test of free fall, the target obstacle tethered in front of the unmanned vehicle falls from a high place according to free fall, quickly approaches the unmanned vehicle equipped with a bionic compound eye microsystem, and triggers a backward

motion. Verification shows that the unmanned vehicle has sensitive backward acceleration, ensuring that it no longer comes into contact with the target during the subsequent swinging process. In the continuous obstacle avoidance test when approaching a target, an obstacle target with initial velocity decelerates and approaches the unmanned vehicle. The unmanned vehicle automatically performs continuous backward actions, always maintaining a safe distance from the target to avoid collision; In the multi-target active obstacle avoidance test, the autonomous vehicle during cruising slows down and turns to avoid obstacles when encountering obstacles. Multiple obstacles in a row can reliably trigger obstacle avoidance actions and ultimately exit towards the direction away from the obstacle target.

The Supplementary Figure 24 is shown below.

REVIEWER COMMENTS

Reviewer #1 (Remarks to the Author):

While it is somewhat regrettable that most responses primarily acknowledged the current system's limitations rather than presenting groundbreaking solutions, we nevertheless confirmed the significance of this research through the appended Supplementary Video. However, we found the discussion regarding the MTF (Supplementary Figure 22) to be inappropriate. Specifically, the presented graph represents the image results of a fabricated microlens array without the integration of sensors, rather than the resolution of the visual system. Given that each microlens is associated with a single pixel of 50 μm , resulting in a Nyquist frequency of 10 lp/mm, the overall system cannot practically achieve the MTF performance indicated. Therefore, the terminology needs to be corrected from "Resolution of the visual system" to "Resolution of the fabricated microlens array".

We sincerely appreciate the reviewer's insightful observation. We have changed the terminology and updated in Supplementary Figure 22, as shown below.

Supplementary Figure 22: Resolution of the fabricated microlens array. MTF represents the resolution of the fabricated microlens array. Observe a specific target card using a lens and read the MTF value, with 0.5 being the resolution contrast optimization point. The maximum resolution obtained from simulation is 416 lp/mm (2.4 μm), while the measured resolution is 3.7 μm , which is better than the 70 μm of the human eye.

Reviewer #2 (Remarks to the Author):

The authors have addressed all questions and it can be published at its current form.

We sincerely thank the reviewer for approving the revision.